Manuscript prepared for Atmos. Chem. Phys.
with version 2015/04/24 7.83 Copernicus papers of the LATEX class copernicus.cls.
Date: 5 October 2017

# The Role of 1D and 3D Radiative Heating in the Organization of Shallow Cumulus Convection and the Formation of Cloud Streets

**Jakub Fabian**[1] and **Mayer Bernhard**[1]

[1]Meteorological Institute, Ludwig Maximilian Universität München, LMU

*Correspondence to:* Jakub Fabian (fabian.jakub@physik.uni-muenchen.de)

**Abstract.** The formation of shallow cumulus cloud streets was historically attributed primarily to dynamics. Here, we focus on the interaction between radiatively induced surface heterogeneities and the resulting patterns in the flow. Our results suggest that solar radiative heating has the potential to organize clouds perpendicular to the sun's incidence angle. To quantify the extent of organization, we performed a high resolution LES parameter study. We varied the horizontal wind speed, the surface heat capacity, the solar zenith and azimuth angles, as well as radiative transfer parameterizations (1D and 3D). As a quantitative measure we introduce a simple algorithm that provides a scalar quantity for the degree of organization and the alignment. We find that, even in the absence of a horizontal wind, 3D radiative transfer produces cloud streets perpendicular to the sun's incident direction, whereas the 1D approximation or constant surface fluxes produce circular, randomly positioned, clouds. Our reasoning for the enhancement or reduction of organization is the geometric position of the cloud's shadow and its corresponding surface fluxes. Furthermore, when increasing horizontal wind speeds to 5 or $10 \, \mathrm{m \, s^{-1}}$, we observe the development of dynamically induced cloud streets. If in addition, solar radiation illuminates the surface beneath the cloud, i.e. when the sun is positioned orthogonally to the mean wind field and the solar zenith angle is larger than $20°$, the cloud-radiative feedback has the potential to significantly enhance the tendency to organize in cloud streets. In contrast, in the case of the 1D approximation (or overhead sun), the tendency to organize is weaker or even prohibited because the shadow is cast directly beneath the cloud. In a land-surface type situation, we find the organization of convection happening on a timescale of half an hour. The radiative feedback, creating surface heterogeneities is generally diminished for large surface heat capacities. We therefore expect radiative feedbacks to be strongest over land surfaces and weaker over the ocean. Given the results of this study we expect that simulations including shallow cumulus convection will have difficulties producing cloud streets if they employ 1D radiative transfer solvers or may need unrealistically high wind speeds to excite cloud street organization.

## 1 Introduction

The advent of airborne and satellite observations allow for a bird's eye view of the atmosphere and, ever since, meteorologists have been fascinated by the striped patterns often evident in cloud systems. Kuettner (1959) presented some early pictures of cloud streets from rocket and aircraft instruments. Descriptions of cloud streets, date back as far as Steinhoff (1935), who gave a detailed description of a long-distance glider flight, or Woodcock (1942) who investigated the soaring patterns of seagulls. Scientific literature documenting the existence and explaining the prerequisites for the formation of cloud streets is plentiful. Brown (1980); Etling and Brown (1993); Weckwerth et al. (1997); Houze Jr (2014) provide a thorough review of past observations and theoretical frameworks. The above literature suggests two prominent effects to be responsible for such vortices, namely inflection-point instabilities (e.g. from cross-roll wind components in a Ekman boundary layer) and thermal instabilities (buoyancy driven). Purely buoyancy driven convection, without any horizontal wind or shear, produces a random pattern of updrafts. Introducing a linear wind shear, the convective elements become stretched out along-wind. Following Grossman (1982): "*At some point (increasing the wind speed/shear) the shearing becomes strong enough so that dynamic instability may interact with buoyancy to produce a*

*hybrid roll vortex/convective cell mechanism. As the shear becomes stronger, shearing instability or roll vortex motion is predominant.*" In this work, we will focus on the radiative impact, with the most prominent effect being cloud shadows which modulate surface fluxes and consequently build up surface heterogeneities. These induced surface heterogeneities are the link between radiative transfer and buoyancy driven convection (Lohou and Patton, 2014; Horn et al., 2015; Gronemeier et al., 2016). Our focus is therefore more on buoyancy driven roll vortices in a linear shear environment (Asai, 1970) and less so on inflection-point instabilities. To that end, we omit cross-wind shear by neglecting Coriolis force and correspondingly neglect the horizontal turning of the wind as it would be the case in an Ekman boundary layer. Several studies investigated the role of surface fluxes on the development of such boundary layer circulations. Here the literature distinguishes between static heterogeneities, i.e. differences in land-surface parameters such as vegetation, surface roughness or surface albedo and dynamic heterogeneities, such as moisture budget or temperature fluctuations. Static heterogeneities in conjunction with shallow cumulus clouds and cloud streets have been examined for example by Avissar and Schmidt (1998); Patton et al. (2005); Rieck et al. (2014). In contrast, Schumann et al. (2002); Wapler (2007); Frame et al. (2009); Gronemeier et al. (2016) investigated the influence of dynamic heterogeneities in surface shading and even considered 3D radiative effects (i.e. the displacement of the shadow). However, they did not include a realistic surface model, but rather adjusted the surface fluxes instantaneously. This does not allow to study the timescales on which radiation and dynamics may interact. Others investigated the influence of shading coupled to an interactive surface model (Vilà-Guerau de Arellano et al., 2014; Lohou and Patton, 2014; Horn et al., 2015). However, one particularly questionable issue with those studies was the application of 1D radiative transfer solvers, which are known to introduce large spatial error in surface heating rates (O'Hirok and Gautier, 2005; Wapler and Mayer, 2008; Wissmeier et al., 2013; Jakub and Mayer, 2015).

Overall, we can summarize that the formation of cloud streets has been extensively explored from theoretical and observational perspectives. The above mentioned studies shed light on the various aspects of interaction with the cloud field but either lack a realistic representation of surface processes, neglect 3D radiative transfer effects or do not examine the relationship concerning the background wind speed.

In this study we strive to overcome these shortcomings and determine the prerequisites for the formation of cloud streets, while our main focus lies on dynamic heterogeneities and (3D) radiative transfer. We try to disentangle the underlying processes with a rigorous parameter study using Large-Eddy-Simulations (LES).

Section 2 briefly outlines the LES model, explains the setup of the simulations and introduces a scalar metric to quantify the organization in cloud streets. In section 3 we interpret the

magnitude of cloud street formations in the parameter space spanning surface properties, background wind speeds and the sun's angles. Section 4 finally summarizes key findings of the parameter study.

## 2   Methods and Experiments

### 2.1   LES Model

The Large-Eddy-Simulations (LES) were performed with the UCLA–LES model. A description and details of the LES model can be found in Stevens et al. (2005). The land surface model included in the UCLA–LES follows the implementation of the Dutch Atmospheric Large-Eddy Simulation code Heus et al. (2010). The simulations presented here use warm micro-physics formulated in Seifert and Beheng (2001) where the formation of rain is turned off to prevent any further complications such as cold pool dynamics. The radiative transfer calculations are performed with the TenStream package (Jakub and Mayer, 2015), which includes a 1D Schwarzschild (thermal only), a $\delta$-Eddington two-stream (solar and thermal), as well as the 3D TenStream (solar and thermal) solver.

The TenStream is a MPI-parallelized solver for the full 3D radiative transfer equation. In analogy to a two-stream solver, the TenStream solver computes the radiative transfer coefficients for up- and downward fluxes and additionally for sideward streams. The coupling of individual boxes leads to a linear equation system which is written as a sparse matrix and is solved using parallel iterative methods from the "Portable, Extensible Toolkit for Scientific Computation", PETSc (Balay et al., 2014) framework. In Jakub and Mayer (2015, 2016), we extensively validated the TenStream by comparison with the exact Monte Carlo code *MYSTIC* (Mayer, 2009).

The most pronounced differences between 1D and 3D radiative transfer solvers, pertaining the setup here, is the displacement of the sun's shadow at the surface. In the case of 1D radiative transfer, the shadow of a cloud is by definition always directly beneath it (so called independent pixel or independent column approximation). Contrarily, 3D radiative transfer allows the propagation of energy horizontally and correctly displaces the clouds shadow depending on the sun's position. The features of 3D radiative transfer in the thermal spectral range are an increased cooling on cloud edges and a smoothed distribution of surfaces fluxes. While we compute thermal radiative transfer in a 3D fashion, we expect these effects to be less important for this setup because feedbacks on the dynamics appear to happen only longer timescales of a day (Klinger et al., 2017) and more importantly because it does not cause any asymmetries in the heating or cooling pattern.

The spectral integration is performed using the correlated-k method following Fu and Liou (1992). The coupling of

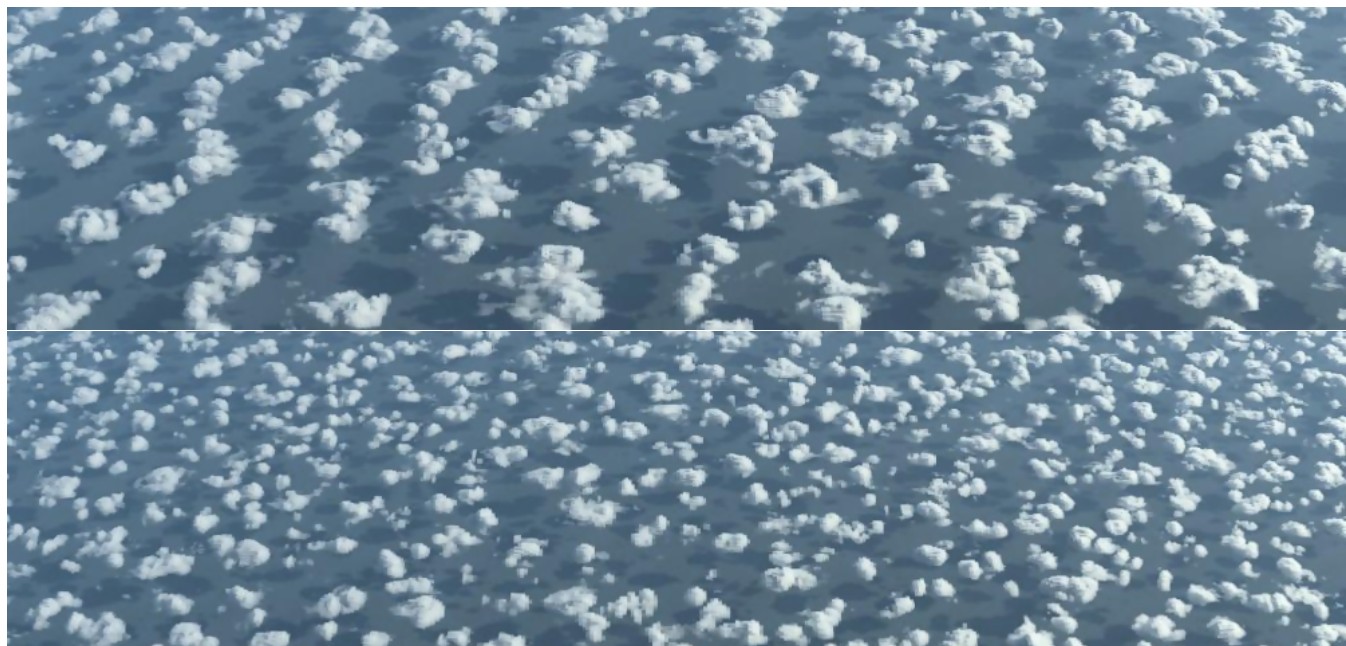

**Figure 1.** Virtual photograph of LES simulations at a cruising altitude of $15\,\mathrm{km}$. Top panel: cloud formation of a simulation driven by 3D radiation (TenStream with sun in the east, i.e. right ($\varphi = 90°$)). Lower-panel: cloud formation of a simulation which was performed with 1D radiation (Two-stream). The specific model setup is the same as referenced in fig. 2, i.e., no background wind and a continental land surface. The simulations differ with respect to cloud size distributions and the organization in cloud streets, the cloud fraction though is the same ($27\,\%$). The visualization was performed with a physically correct rendering with MYSTIC (MonteCarlo solver in libRadtran (Mayer, 2009; Emde et al., 2015)).

the TenStream solver to the UCLA–LES follows the description in Jakub and Mayer (2016). One exception is the use of the Monte-Carlo-Spectral-Integration (Pincus and Stevens, 2009) which we do not use because of limitations with regards to computations involving interactive surface models (Pincus and Stevens, 2013).

## 2.2 Model Experiment Setup

The base setup of the UCLA–LES simulates a domain of $50\,\mathrm{km} \times 50\,\mathrm{km}$ with a horizontal grid length of $100\,\mathrm{m}$ and $50\,\mathrm{m}$ vertically. The simulations start from a well-mixed initial background profile with a constant virtual potential temperature ($292\,\mathrm{K}$) in the lower $700\,\mathrm{m}$ and increases by $6\,\mathrm{K\,km^{-1}}$ above. Water vapor near the surface amounts to $9.5\,\mathrm{g\,kg^{-1}}$, decreasing with $-1.3\,\mathrm{g\,kg^{-1}\,km^{-1}}$. The surface model has four layers which have the same initial temperature of $291\,\mathrm{K}$, are stripped of vegetation and are soaking wet (saturated clay with $30\,\%$ water volume mixing ratio). The surface albedo for shortwave radiation is set to $7\,\%$. The land-surface model solves the surface energy balance equation for an imaginary skin layer which often has no heat capacity. We vary the heat capacity of the surface skin layer $C_{\mathrm{skin}}$ to mimic a water layer covering the surface. The heat capacities are chosen to be representative for situations ranging from continental land surfaces to a well mixed ocean. The

thickness of this imaginary water layer lends the simulations and the radiative transfer a memory on the surface. All other parameters of the land-surface model such as surface resistances or roughness lengths for momentum or heat are kept constant in order to focus on these memory effects.

The focus of this study is to determine the interplay of radiation with the atmosphere, the surface and the clouds, and finally take a closer look on the formation of cloud streets. To that end we run the simulations with five free parameters, namely the heat capacity of the surface skin layer ($C_{\mathrm{skin}}$), the background wind ($\boldsymbol{u}$, i.e. west-winds), the solar zenith ($\theta$) and azimuth ($\varphi$) angle as well as with different radiative transfer approximations (see table 1). The coupling of radiative transfer to the land-surface model is realized in four ways. We either compute the net surface irradiance $Q_{\mathrm{net}}$ with a 1D $\delta$-eddington two-stream solver , or employ the 3D Ten-Stream solver, with two azimuth angles. Additionally, we conducted the experiments where $Q_{\mathrm{net}}$ is set to a prescribed constant value (spatial and temporal average of the surface irradiance of the corresponding 1D simulation).

The time it takes the simulations to form the first clouds depends on the choice of the parameters. Foremost the solar zenith angle determines the energy input into the atmosphere and the surface (lower positioned sun hence leads to a later onset of cloud development). To compare the heterogeneous simulations we limit the following analysis to the

**Table 1.** Parameter space for the LES simulations: the mean west wind $\boldsymbol{u}$, the solar azimuth and zenith angle $\varphi,\theta$, the surface skin heat capacity $C_{\mathrm{skin}}$ as a water column equivalent and three settings for the computation of net radiative surface fluxes ($Q_{\mathrm{net}}$). The radiative transfer computations are done either with a 1D $\delta$-Eddington two-stream , with the 3D TenStream solver or simulations with constant mean net irradiance. Variations of the solar azimuth $\varphi$ are only applied for 3D radiative transfer. Values of $Q_{\mathrm{net}}$ in case of simulations without interactive radiative transfer were set to the mean surface irradiance of the 1D simulations. In total a number of 192 simulations.

| $\boldsymbol{u}$ | 0, 5, 10 | $\mathrm{m\,s^{-1}}$ |
|---|---|---|
| $\varphi$ | 90, 180 | $^{\circ}$ |
| $\theta$ | 20, 40, 60, 75 | $^{\circ}$ |
| $C_{\mathrm{skin}}$ | 1, 10, 100, 1000 | cm |
| $Q_{\mathrm{net}}$ | constant, 1D, 3D | |

time-steps (output every $5\,\mathrm{min}$) where the cloud fraction is between $10\,\%$ and $50\,\%$ (typical for shallow cumulus convection,e.g. Seifert and Heus (2013)). Most simulations produce clouds after about one hour and show an increase in cloud cover up to and beyond $50\,\%$ in the first $6\,\mathrm{h}$. Simulations with low positioned sun took longer and were hence run for a longer period of $12\,\mathrm{h}$. Our analysis is mostly independent of the specific, individual course of each simulation as we find robust signals across the various groups of parameters. The interested reader, however, is referred to Jakub (2016, sec. 3.2) for further details (e.g. liquid water path, cloud fraction, mean cloud size distribution) on the evolution of a typical simulation.

Figure 1 shows a photo rendering of the LES cloud field for two simulations with differing options for the radiative transfer solver. In the top panel, 3D radiative transfer is considered with the sun positioned in the east (zenith $\theta = 60°$) and in the bottom panel panel 1D solver is applied where the shadow is by definition always cast directly beneath the clouds. In the former the organization in cloud streets perpendicular to the sun's incident angle is evident whereas the latter (1D) does not seem to organize in any way. Figure 2 presents the liquid water content and the surface heat flux for the same two simulations plus one 3D simulation where the sun is in the south. This time we look at volume rendered liquid water content and surface heat fluxes for the full domain. In figs. 1 and 2, simulations with 3D radiative transfer show organization in cloud streets with length scales of up to $20\,\mathrm{km}$, perpendicular to the sun's incident angle. We can clearly identify these coherent cloud structures with the naked eye. However, to solidify our claims, we present a quantitative measure for the cloud distribution.

## 2.3 Correlation Ratio

Since we do not deal with towering and tilted or multilayer clouds we can use the cloud mask as a proxy to separate individual clouds. We derive the cloud mask as the binary field of the liquid water path (LWP $> 0$). We then use the normalized 2D auto correlation function of the cloud mask to analyze the spatial distribution of cloudy and clear-sky patches. The three upper panels of fig. 3 illustrate the 2D correlation coefficient for the three simulations presented in fig. 2.

Next, we use the transects of the correlation coefficient along the x- and y-axis (indicated as a black line). The lower panels in fig. 3, respectively, show the linearly interpolated line-cuts of the discrete auto-correlation function. The location where the normalized correlation coefficients goes to zero defines the mean distance from a cloudy pixel where it is more likely to find a clear-sky pixel. We use the north-south and the east-west distances $d_{\mathrm{NS}}$ and $d_{\mathrm{EW}}$, respectively, to define the correlation ratio $R_{\mathrm{c}}$ as:

$$R_{\mathrm{c}} = d_{\mathrm{NS}}/d_{\mathrm{EW}}$$

This definition would miss cloud streets in diagonal direction which, however, is no limitation for our analysis. For one, we know that the background wind induces cloud streets along the mean wind direction, i.e. here in the west-east component (see e.g. Weckwerth et al. (1997)). At the same time we hypothesize that radiatively induced effects will be either along or perpendicular to the incident solar beam, i.e. follow the surface inhomogeneities (see, e.g., Gronemeier et al. (2016). The two major directions should therefore capture the dominant effects of dynamically and radiatively induced cloud dynamics.

The correlation ratio reduces a cloud field snapshot into a scalar which yields $R_{\mathrm{c}} = 1$ for symmetrically distributed clouds, $R_{\mathrm{c}} < 1$ for organized cloud fields along the north-south direction and $R_{\mathrm{c}} > 1$ if cloud features are arranged east to west.

## 3 Results and Discussion

As an example for the evolution of convective organization, fig. 4 illustrates the correlation ratio $R_{\mathrm{c}}$ over time for one of the earlier introduced simulations (depicted in fig. 2). In the simulation, first cumulus clouds occur after about half an hour with the clouds being oriented randomly. The resulting shadowing of these clouds introduces surface temperature heterogeneities which in turn act on the flow through changes in latent and sensible heat fluxes. About one hour after the onset of clouds, we find the convection to organize into bands from north to south ($R_{\mathrm{c}} < 1$). To further highlight the involved timescales, we restart the simulation from $2\,\mathrm{h}$ onwards with a 90° rotated sun and find that convection changes from a north to south orientation to bands from east to west in approximately one hour. This example yields a $1/e$

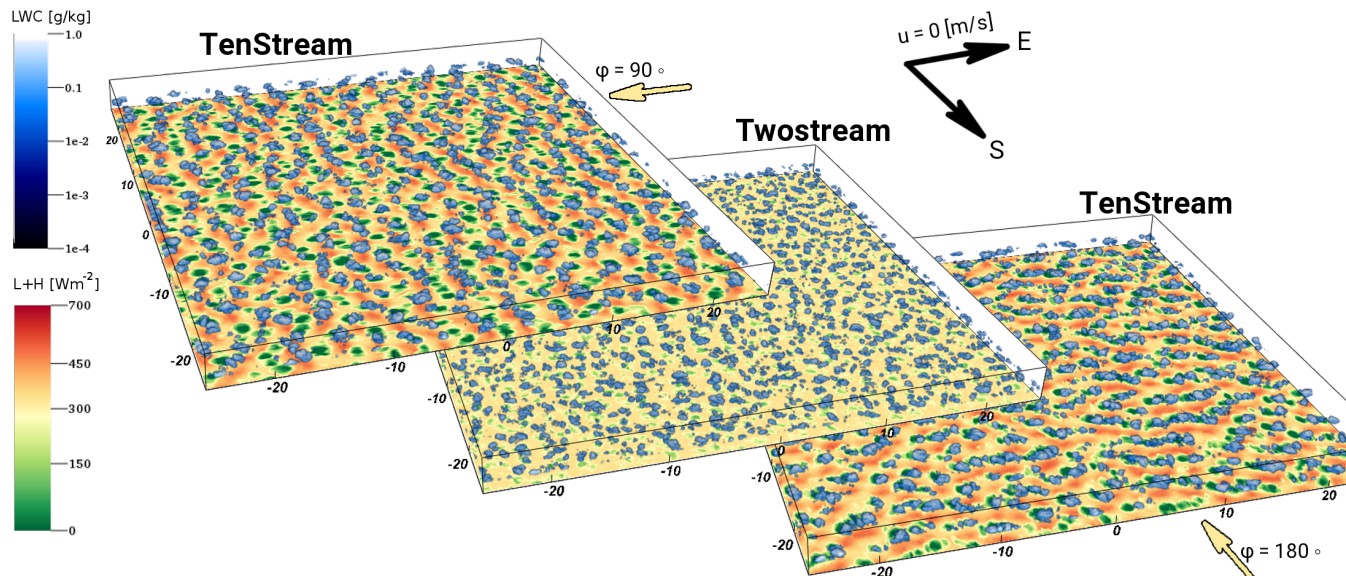

**Figure 2.** Volume rendered liquid water mixing ratio (LWC) and surface latent and sensible heat flux $(L + H)$ for three simulations. The cloud scene of the left and mid panel have already been presented in fig. 1. In the left panel, radiative transfer calculations are performed with the TenStream solver and the sun is positioned in the east $(\varphi = 90°)$. The simulation in the mid-panel is driven by a 1D two-stream solver, whereas the right panel simulation also employs the TenStream solver but the sun shining from the south $(\varphi = 180°)$. The solar zenith angle is in all three simulations $\theta = 60°$, the mean background wind speed is $0\,\text{m s}^{-1}$ and the surface skin heat capacity set to an equivalent of $1\,\text{cm}$ water depth (representative for continental land surface). The snapshot shows the simulations after $3\,\text{h}$ model time at a cloud fraction of $27\,\%$. Volume rendered plots were created with VISIT (Childs et al., 2012).

timescale for convective organization of half an hour. This timescale will however, depend on several factors, most certainly on the solar zenith angle and the surface heat capacity which determine the timescales at which surface heterogeneities can be introduced.

To reduce the information of convective organization into a single scalar value, we compute the mean correlation ratio $\overline{R_c}$ as the arithmetic mean of $R_c$ calculated at all output time-steps (every 5 minutes) where the cloud fraction is between $10\,\%$ and $50\,\%$. The aim of the cloud fraction filtering is to allow a comparison of simulations with varying temporal evolutions due to different energy inputs (solar zenith angles) and heat sinks ($C_{\text{skin}}$).

The basis for the following analysis is the evaluation of mean correlation ratios as a function of the five free parameters, $u$, $\varphi$, $\theta$, $C_{\text{skin}}$, and the radiative transfer solver (for details, see table 1). Figure 5 shows the mean correlation ratio $\overline{R_c}$ for each of the 192 simulations. The three panels show results for different horizontal background wind speeds, $0\,\text{m s}^{-1}$, $5\,\text{m s}^{-1}$ and $10\,\text{m s}^{-1}$. Each panel's x-axis is divided into four categories for the surface skin heat capacity and the colorbar describes the solar zenith angle. Additionally, four different markers denote the various options concerning the radiative transfer solvers while the rotation of triangle markers (3D RT) denote the azimuth angle.

We will first focus on the left panel which shows the correlation ratios for the simulations without any background wind and later move on to simulations with wind. In other words, we start by focusing on purely radiative effects and their influence on the organization of convection and eventually add dynamically induced cloud streets to the discussion.

## 3.1   Without Wind: $u = 0\,\text{m s}^{-1}$

The three simulations presented in section 2 are located on the far left panel of fig. 5 with a surface skin heat capacity equivalent of $1\,\text{cm}$ water column (furthest to the left shaded area). Correspondingly, the markers for 3D radiative transfer are shown as triangle markers in light blue (zenith angle of 60°). The upward triangle represents the sun positioned in the south and yields a mean correlation ratio of 1.5 (rolls produced west to east). In contrast, the left rotated triangle presents a sun positioned in the east and shows a mean correlation ratio of .7 (rolls produced south to north). The simulation with 1D radiative transfer is presented with a diamond shaped marker and shows a mean correlation ratio of $\approx 1$ (no organization).

To explain the concept of why 3D RT creates rolls, we may setup a quick thought experiment. First start with the assumption that there already is a single cloud which will cast the shadow along the sun's incident angle. The surface fluxes for latent sensible heat will be smaller in the shadowy area and hence we expect the next convective plume to rise in sun-lit areas. Figure 6 illustrates the concept for

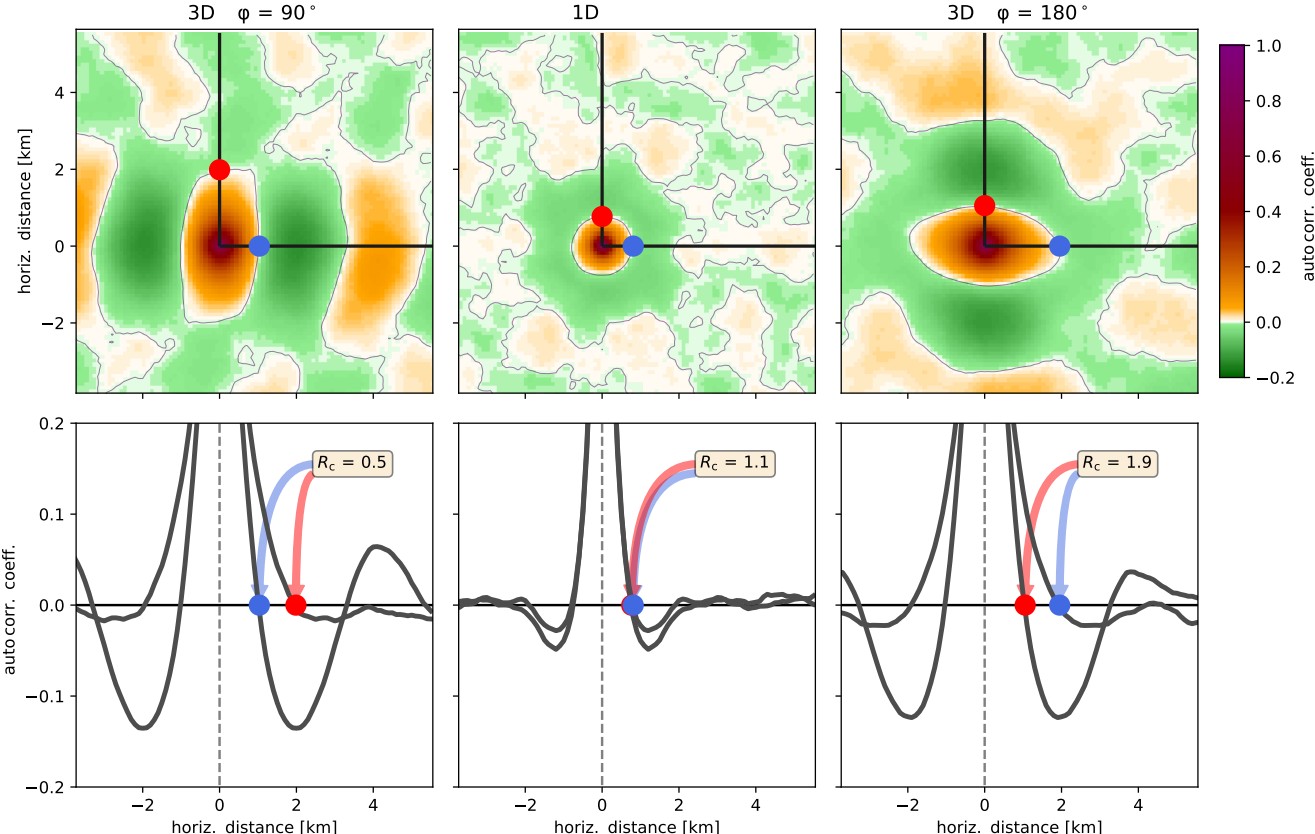

**Figure 3.** The panels exemplarily depict the auto-correlation coefficients of the cloud distribution in the three simulations presented in fig. 2. The upper panels show the normalized 2D autocorrelation coefficient with two intersection lines in the North-South (vertical) and the East-West (horizontal) direction. The markers pinpoint the distance in N-S (red) and E-W (blue) direction, respectively, where the auto-correlation coefficient reaches a zero value and therefore denote the distance where it becomes more likely not to find a cloud. The lower panels follow the black line-cuts and further describe the two transects depicting the correlation function's root points from which we derive the correlation ratio. Simulations with 3D radiative transfer (left and right panels) shows in contrast to 1D radiative transfer (mid panel) a distinct asymmetry perpendicular to the solar incidence angle. The organization of clouds and their alignment is represented in values of the correlation ratio $R_c$ being less than or greater than one for alignment along the y- or x-axis, respectively.

a single cloud and the resulting pattern for surface fluxes. The schematic only constrains convection to be less favorable on the shadowy side but it does not necessarily favor the perpendicular directions over the direction towards the sun. However, if a cloud would evolve on the sun-facing side, the resulting shadow would in turn lead to a faster dissipation of the initial cloud and is thereby an unstable environment for persistent cloud patterns. Following this, we expect the convection to occur favorably perpendicular to the sun's incident angle purely from geometric reasoning.

It is also clear from the horizontal axis of fig. 5 that higher heat capacities lead to less pronounced formations of cloud streets which is to be expected because it weakens the radiative impact and consequently reduces the dynamically induced surface heterogeneities. Yet, though weaker, we still find an impact in 3D radiative transfer simulations even for a water column equivalent of 10 m. In this case with such high

surface heat capacities, the simulations do not exhibit any variability in surface fluxes and radiation solely acts through atmospheric heating. We recover this behavior also in simulations with a fixed sea surface temperature or with constant latent and sensible surface fluxes (not shown). In Jakub (2016, fig. 3.22), we show that the asymmetric heating of the cloud sides (or similarly in Wapler (2007); Gronemeier et al. (2016) for displaced surface shadows) introduces a secondary circulation by lifting the sun-lit side and enhancing subsidence on the shadowy side. This asymmetry introduces a wind shear component consisting of a horizontal wind away from the sun at cloud height and towards the sun near the surface. Given that the effects of atmospheric heating is much smaller and happens on longer timescales compared to the surface feedback we put the interpretation aside for another time.

Simulations with one-dimensional radiative transfer or con-

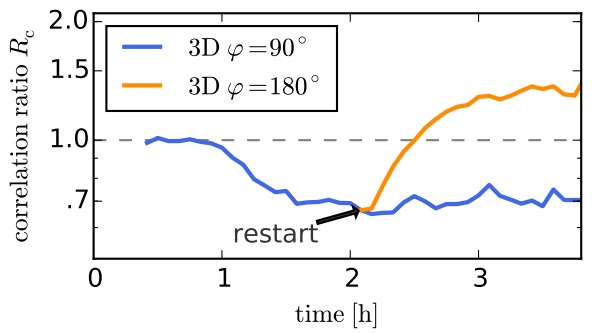

**Figure 4.** Time evolution of the correlation ratio $R_c$ (e.g. as in fig. 2). The solar zenith angle is $\theta = 40°$, there is no mean background wind speed ($u = 0\,\mathrm{m\,s^{-1}}$) and the surface skin heat capacity is set to an equivalent of $1\,\mathrm{cm}$ water depth (representative for continental land surface). The radiative transfer is computed with the TenStream solver and the sun is positioned in the east ($\varphi = 90°$) The first shallow cumulus clouds develop with a random orientation ($R_c = 1$). The radiative response (i.e. surface shadows) changes the organization of convection to bands from north to south $R_c < 1$ in about one hour. Additionally, to examine the timescales of radiatively induced organization of convection, we perform a restart of the simulation with the sun positioned in the south ($\varphi = 180°$). Once the sun is rotated, it takes the simulation again about one hour to change the orientation of convection into bands from east to west ($R_c > 1$).

stant $Q_{\mathrm{net}}$ do not produce cloud streets which is reflected by correlation ratios $R_c \approx 1$. If we apply the same geometric reasoning from fig. 6 for these simulations, where the shadow is either directly beneath the cloud or with no heterogeneity at all, it is clear that there can be no preferential direction for convective organization.

Three-dimensional radiation calculations with high or low solar zenith angles also show a reduced production of cloud streets. This is, (a) because low zenith angles (sun above head) practically behave just as 1D radiative transfer, and (b), because large zenith angles (low sun, smaller heating rates) have a weaker potential to create surface heterogeneities.

### 3.2 Medium Wind: $u = 5\,\mathrm{m\,s^{-1}}$

The middle panel of fig. 5 presents the correlation ratios for simulations with a horizontal background wind of $5\,\mathrm{m\,s^{-1}}$. If we first shift our attention to the simulations with constant surface irradiance $Q_{\mathrm{net}}$ (round markers), it is evident that the introduction of a mean wind profile leads to the formation of cloud streets ($\overline{R}_c > 1$), irrespective of radiatively induced surface heterogeneities. The fact that we find cloud streets also without any radiation is not surprising and is expected from the literature on the formation of buoyancy driven cloud streets (introduced in section 1. Furthermore, we find a spread in the development of cloud streets depending on the magnitude of the prescribed $Q_{\mathrm{net}}$, with cor-

relation ratios ranging from 1 to 5. The fact that buoyancy driven cloud street organization is favored in slightly unstable conditions (low sun) compared to stronger instabilities (high sun) agrees well with observations (e.g. Woodcock (1942); Priestley (1957); Grossman (1982); Weckwerth et al. (1997)).

So far we discussed only the simulations with constant $Q_{\mathrm{net}}$. When we look at land surfaces that are coupled to radiative transfer calculations (1D and 3D markers in fig. 5), we find that radiative heating may either enhance the organization ($\overline{R}_c$ up to 13) or counter-act it ($\overline{R}_c < 1$). The following paragraph examines the superposition of dynamically and radiatively induced tendencies to organize the clouds.

Let's consider the case that there is a dynamically induced cloud street along the mean background wind, i.e. from west to east. Quasi 1D radiation (1D and 3D if sun is close to zenith) casts a shadow onto the cloud's updraft region and therefore hinders further development of the cloud. This results in values for the correlation ratio of $\overline{R}_c \approx 1$. Similarly, 3D radiation where the azimuth is in the same direction as the wind (here east, $\varphi = 90°$, left-rotated markers) also inhibits the formation of cloud streets or may even oppose the dynamically induced organization and produce correlation ratios $\overline{R}_c < 1$.

In contrast, for 3D radiative transfer with solar incidence perpendicular to the mean wind, i.e. sun from south or north, and permitted that the sun's zenith angle allows to illuminate the surface beneath the cloud ($\theta > 20°$), we find that the radiative tendency to organize the clouds amplifies the dynamically one. This synergistic behaviour results in correlation ratios $\overline{R}_c$ between 5 and 13.

As mentioned previously in section 3.1, we again find a generally diminished influence of surface radiative heating in simulations with larger surface heat capacities.

### 3.3 Strong Wind: $u = 10\,\mathrm{m\,s^{-1}}$

A stronger background wind profile of $10\,\mathrm{m\,s^{-1}}$ principally shows similar behavior as the case that was presented with medium wind speeds (see right panel of fig. 5). The mean correlation ratios of purely dynamically induced cloud streets (simulations with constant $Q_{\mathrm{net}}$, i.e. circle markers) cover an increasingly large range of ratios from 2 to 14. Strong solar radiation coupled with small surface heat capacities still manage to efficiently suppress the formation of cloud streets (i.e. $\overline{R}_c$ consistently smaller than purely dynamic values). Whereas illumination perpendicular to the wind direction ($\varphi = 180$ and $\theta > 20°$) again greatly amplifies the occurrence of cloud streets. This might be surprising if we consider that horizontal wind should indeed smooth out the impact of radiative surface heating. Lohou and Patton (2014) for example also suggest that wind speeds of $10\,\mathrm{m\,s^{-1}}$ may decouple the effects of dynamically induced surface heterogeneities from the evolution of clouds. However, if we consider that the dynamically induced cloud streets have typ-

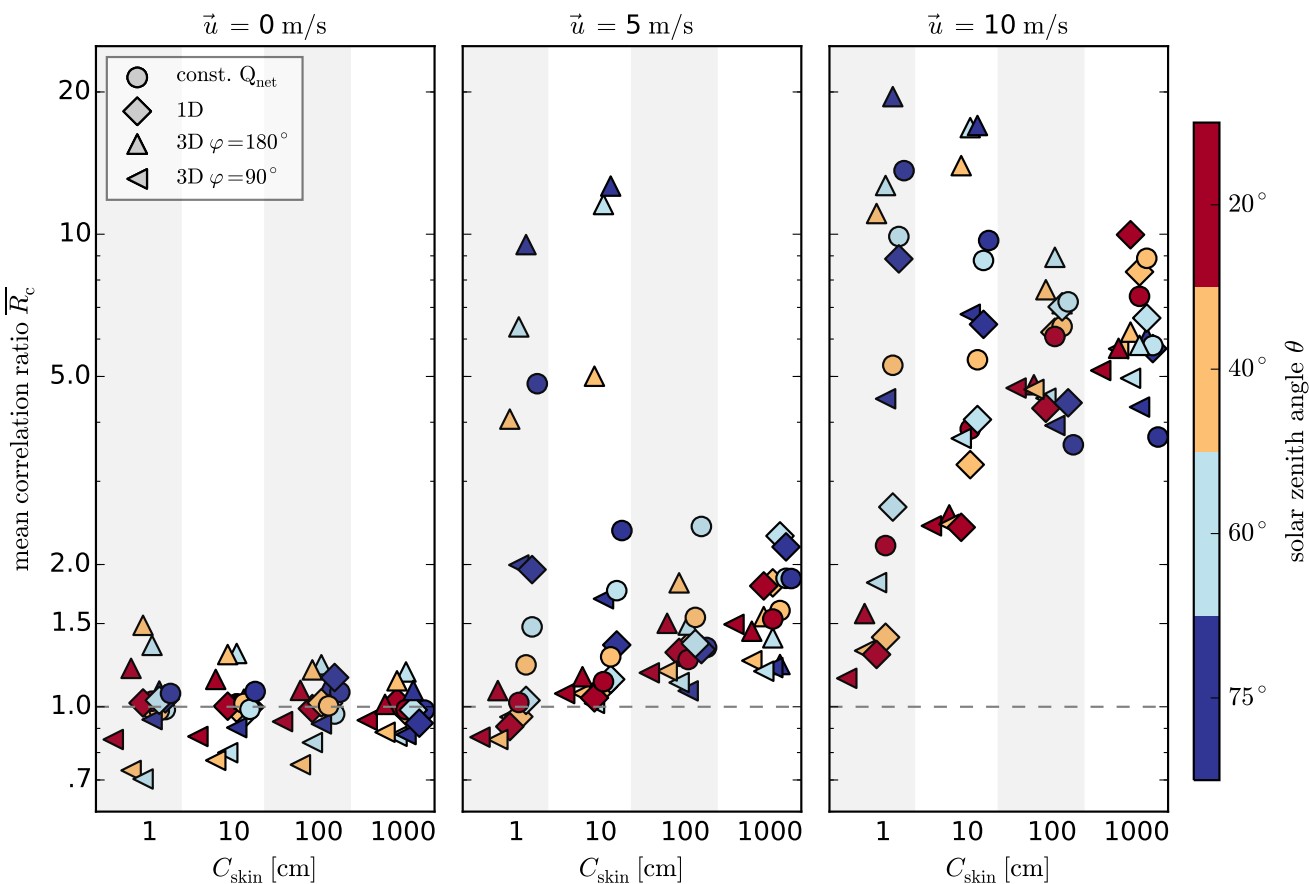

**Figure 5.** Correlation ratio for simulations with a variable surface skin heat capacity ($C_{\text{skin}}$), solar zenith angle ($\theta$), and three wind veloc­ities (panels left to right). Shaded areas group simulations with a constant $C_{\text{skin}}$ according to their respective values, while the horizontal spread inside a group is merely to separate data-points visually. Wind-component $u$ is always from west to east while the individual markers denote simulations where the surface irradiance $Q_{\text{net}}$ is set to a constant value, or is computed either with a 1D two-stream solver, or with the 3D TenStream where the sun is either shining from the south ($180°$) or from the east ($90°$). The correlation ratio is averaged over all time-steps where the cloud fraction is between $10\,\%$ and $50\,\%$.

ical length scales of $50\,\text{km}$ (Kuettner, 1959), then, as far as radiative heating at the surface is concerned, the cloud ap­pears to be standing still. In other words, when a dynami­cally induced cloud feature aligns in such a way that it per­sistently shades a surface region for an extended period of time, we expect that the radiatively induced surface hetero­geneities in turn interact with the flow. It is this intricate linkage between dynamically induced cloud structures and (3D) radiative transfer that may enable or prohibit the forma­tion of cloud streets.

## 4   Summary & Conclusions

The formation of cumulus cloud streets was historically attributed primarily to dynamics. This work aims to doc­ument and quantify the generation of radiatively induced cloud street structures. To that end we performed 192 LES

simulations with varying parameters (see table 1) for the horizontal wind speed, the surface heat capacity, the so­lar zenith and azimuth angle, as well as for different radia­tive transfer solvers (section 2.2). As a quantitative measure for the development of cloud streets, we introduce a sim­ple algorithm using the autocorrelation function on the cloud mask (section 2.3), which provides a scalar quantity for the degree of organization in cloud streets and the alignment along the cardinal directions.

We find that, in the absence of a horizontal wind, 3D ra­diative transfer produces cloud streets perpendicular to the sun's incident direction whereas the 1D approximation or constant surface irradiance produce circular, randomly po­sitioned, clouds. Our reasoning for this is the geometric po­sition of the cloud's shadow and the corresponding feedback on surface fluxes which enhances or diminishes convective tendencies (see fig. 6 for details). While the simulations indi­cate that there exists an influence due to atmospheric heating

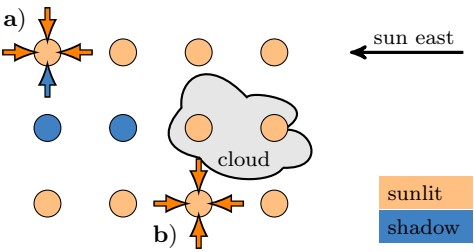

**Figure 6.** Sketch from an aerial view depicting surface fluxes in the vicinity of a cloud with a tilted solar incidence. The cloud casts a shadow on the westward surface pixels (blue dots). The available convective energy is directly proportional to latent and sensible heat release of the surface in the vicinity of the convective updraft. Arrows illustrate the confluence of near surface air masses from adjacent pixels in a thermally driven updraft event. Convective tendencies will be weaker on pixels that are adjacent to shaded patches, e.g. at $a$). In contrast, pixels that are surrounded by sun-lit patches, e.g. $b$), are likely to show enhanced convective motion. This pattern favors the organization of cumulus convection in stripes perpendicular to the sun's incident.

rates, we find that the differences between 1D and 3D radiation stem predominantly from surface heating, i.e. the horizontal displacement of cloud shadows. Furthermore, with increasing horizontal wind speeds of 5 or $10\,\mathrm{m\,s^{-1}}$, we observe the development of dynamically induced cloud streets. The dynamical formation of cloud streets is not particularly surprising, but leads to the question if and how radiative transfer interacts with the organization of convection.

We find that if solar radiation illuminates the surface beneath the cloud, i.e. when the sun is positioned orthogonal to the mean wind field and the solar zenith angle is larger than $20°$, the cloud-radiative feedback may significantly enhance the tendency to organize in cloud streets. In contrast, in the case of the 1D approximation (or also 3D if the sun is aligned with the mean wind), the tendency to organize in cloud streets is weaker or even prohibited because the shadow is cast directly beneath the cloud, weakening the cloud's updraft. The timescale of the convective organization through radiative transfer is found to happen on the order of one hour (see fig. 4). The radiative feedback, creating surface heterogeneities is generally diminished for large surface heat capacities. We therefore expect radiative feedbacks to be strongest over land surfaces and less so over the ocean. Given the results of this study we expect that simulations including shallow cumulus convection will have difficulties to produce cloud streets if they employ 1D radiative transfer solvers or, may need unrealistically high wind speeds to excite cloud street organization.

An interesting future topic would be the influence atmospheric heating rates on the evolution of cloud shapes, particularly the corresponding timescales and how the introduced asymmetry and shear changes the local flow. Mov-

ing forward, we will examine if the relationship between radiative transfer and convective cloud streets also applies to the real world with all the complexities of a diurnal cycle or static surface heterogeneities combined with complex wind fields. Several studies perform detailed analyses on the footprint of static surface heterogeneities in windy conditions, i.e. how upstream heterogeneities influence the characteristics of boundary layer dynamics (e.g. (Raasch and Harbusch, 2001; Prabha et al., 2007; Courault et al., 2007; Maronga and Raasch, 2013; Chen et al., 2015; Gronemeier et al., 2016)). It may very well be worth to revisit their analyses and particularly focus on static and dynamic(radiative) heterogeneities. A promising start is an analysis of the simulations within the HDCP$^2$ project (Heinze et al., 2017) which will allow us to test the here proposed interpretations in a more realistic setup.

## 5 Code availability

The UCLA–LES model is publicly available at https://github.com/uclales. The calculations were done with the modified radiation interface which is available at git-revision "56587a6".

To obtain a copy of the TenStream code, please contact one of the authors. This study used the TenStream model at git-revision "5e0a2d5". For the sake of reproducibility we provide the input parameters for the here mentioned UCLA–LES computations along with the TenStream sources.

*Acknowledgements.* This work was funded by the Federal Ministry of Education and Research (BMBF) through the High Definition Clouds and Precipitation for Climate Prediction (HD(CP)2) project (FKZ: 01LK1208A, 01LK1507D). Many thanks to Cathy Hohenegger, Bjorn Stevens and the DKRZ, Hamburg for fruitful discussions and for providing us with the computational resources to conduct our studies. Special thanks are also due to Alois Dirnaichner and the anonymous reviewers who improved this manuscript by proofreading and commenting on the original manuscript.

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
