# Peer review of "The Role of 1D and 3D Radiative Heating in the Organization of Shallow Cumulus Convection and the Formation of Cloud Streets"

_Atmospheric Chemistry and Physics, 2017_

## Referee Comment (RC1) · Anonymous Referee #1 · 27 May 2017

This paper discusses results from a series of LES simulations of shallow nonprecipitating convection focusing on the role of 3D radiative transfer and the coupling with the surface processes. Overall, I feel this is a nice study and it should be eventually published. However, he analysis is very superficial and does not really give justice to the tremendous amount of computation that went into producing the dataset. I provide general comments below where some suggestions of additional analysis are given, and subsequently follow with detailed specific comments.

General comments:

1. One of key aspect of the roll-type shallow convection is the presence of low-level shear associated with the Ekman boundary layer. This is really not mentioned in the

introduction and I feel this is an essential omission. I think the shear explains the key impact of the mean wind as documented in Fig. 4. I have more points on that aspect in the specific comment section below, but a discussion of numerical studies (starting with Mason and Sykes QJ 1982, p. 801) as well as observational studies have to be brought in the revision (Weckwerth at el. is just mentioned in passing without any reference to the dynamics). There is also a wealth of theoretical studies on the stability of shear flows in unstable stratification focusing on the development of roll-type circulations, starting with Asai (JMSJ 1970, p. 129). I understand that the authors specialize in the radiative transfer and not in the atmospheric dynamics, but the poor treatment of the dynamical aspects needs to be corrected. My suggestion is the authors trace back citations to the papers listed above and provide an appropriate discussion on the role of boundary-layer shear in determining the organization. Overall, I feel the dynamics is the key, and radiation provides just a small (although quite interesting!) modification. But I feel that unequivocally separate the two is difficult.

2. The model setup is described with insufficient detail. For instance, sending the reader to the description of the land surface model in Heus et al is not appropriate. The C_skin parameter in Table 1 is not explained and I did not know what it really meant. In the discussion of model results this becomes obvious: this is the depth of the well-mixed layer of water that responds to radiative and surface heat fluxes. This is critical to the specifics of the simulation as the shadow on the surface is only important through its effect on the surface sensible and latent fluxes, doesn't it? Ocean response to the shadow can be argued to be quite small at spatial and temporal scales this study is concerned with, whereas land surface would respond quite rapidly. Similarly, significant wind moves the cloud and its shadow, and the surface may not have time to respond. These aspects of the model need to be presented in detail so the reader is aware of the surface response in various simulations. Are the surface momentum fluxes (i.e., surface friction) included in the model setup? If so, what is the surface drag (or whatever parameter is used to describe surface roughness) for the momentum? For instance, to mimic the difference between land-surface (small C_skin) and the ocean

(large C_skin), the surface drag should also be changed (larger over land and smaller over the ocean). This aspect should be at least mentioned in the description of the model as the surface drag affects the shear across the boundary layer.

3. I would like to see more analysis of bulk properties of the cloud field to put model results into perspective. For instance, the authors should show evolution of the cloud cover for various simulations, BL depth (differences in the cloud mean size evident in Fig. 1 suggests to ma that BL is deeper in the upper panel as clouds seem larger), depth of the cloud field, wind profiles across BL in various simulations, etc. etc. Differences in those bulk properties can affect organization of shallow convection as well and better isolating them from the effects of 3D radiation would be desirable. At the moment, the authors provide very speculative discussion of the model results (see specific comments) and I think some of the bulk differences may be used to better explain the results as well.

4. As far as I can tell, shallow convection organization develops gradually and the time scale is relatively long (hours; this can also be better quantified in the analysis). In nature, the sun is moving around, so both the azimuth and zenith angles are slowly changing. So the idealized setup may be questioned if one has to wait long time for the organization to develop. This aspect needs at least to be recognized in the manuscript.

Specific comments:

1. The title needs revision. "The role"... "on" is not correct. "In" would be better, but replacing "role" with "impact" would be more appropriate.

2. L. 92: "resolution" has to be replaced with "grid length".

3. L. 96: I do not understand "...layers of the surface model are soaking (30% vmr)". Please rephrase. Is the Bowen ratio the same in all simulations? This affects buoyancy flux that drives the boundary layer dynamics.

4. L. 111. Lower sun means lower energy input, hence later convection development,

correct? I would also think that this leads to different evolution of the boundary layer depth, an aspect that might be important as well.

5. I suggest adding a table with simulation acronyms and apply them throughout the text for an easy reference.

6. Do various simulations have different destabilization rates across the lower troposphere? This may have some impact on convection as well. See major point 3 above.

7. Fig. 2 is too small. Consider splitting into separate figures or use vertically-stack panels.

8. L. 136. Is this the wind direction, or shear? What is "along track"?

9. I feel Fig. 4 is the key result of the study. But some aspects are really not mentioned in the discussion. i) The spread between simulations with different $C_{skin}$ narrows with the $C_{skin}$ increase. Does this suggest some dynamical effects through surface fluxes? ii) The correlation ratio is much larger for the strong wind case, no doubt because of the role of Ekman shear across the boundary layer.

10. I found the discussion in section 3 speculative and not supported by the analysis. For instance, Fig 5 can be supported by the analysis of model data. That said, my problem is that changes in the surface fluxes do not translate immediately into changes of the boundary-layer structure. The argument is likely correct for the surface layer, but I am not sure how rapidly these changes are passed higher up. Another aspect is the role of secondary circulations that can either support or suppress development of roll-type convection. The discussion on lines 185-190 seems to suggest that the authors think this happens, but I suggest using model data in an attempt to document that. For instance, are there any systematic differences in the updraft/downdraft structure between sunlit and shadow part of the cloud? One should investigate that.

11. How the wind (and thus the boundary layer shear) is maintained? Again, major point 2 above.

12. L. 210 – 217. Can these speculations be supported by appropriate analysis of model data (e.g., shear, boundary layer depth, etc).

13. Suggestion for the future: one can apply different surface roughness to explore the impact of shear. Also, one can vary Coriolis parameter (including a change of sign to mimic the southern hemisphere) to better separate dynamical and radiative effects.

14. L. 245-250: I am sure there are more recent references that show observational estimates of the relevant scales than Kuettner 1959.

15. I found the conclusion section too brief and not providing the justice to the wealth of results the authors have. In particular, dynamical aspects are really not discussed at the appropriate detail level throughout the text and thus in the summary section.
* * *

---

## Referee Comment (RC2) · Anonymous Referee #2 · 16 Jun 2017

The manuscript explores the role of 3-D radiative transfer, particularly in terms of its creation of surface shadowing, on cloud organization into streets. It employs a large number of LES simulations for different solar configurations, surface heat capacities, and horizontal wind speeds to evaluate the contribution of 3-D radiation to cloud street formation. It is found that even in the absence of horizontal wind, 3-D radiation has a tendency to generate organization of streets orthogonal to the solar azimuth. In the presence of horizontal wind, 3D radiation can either enhance or suppress the tendency to organization into streets depending on the configuration.

I found the manuscript intriguing and highly interesting, but in need of quite a few

clarifications, possible analyses to dig a little deeper, and technical corrections (grammatical). (I will address the former and leave the grammatical aspects to a technical editor.)

1) The paper is very short and could do with more material. To start with it should inform the reader about the theory of streets. Line 51/52 is insufficient. The authors tend to be in a hurry to tie the paper up and not deal with details like teasing out the extent to which horizontal photon transport contributes to the results (Line 190). I would have appreciated more analysis. A few choice simulations to focus on various issues would greatly add to the impact of the paper.

2) The influence of 3-D longwave cooling should be discussed.

3) I liked the intuitive sketch (Fig. 5) but would appreciate a similar sketch pertaining to the dynamics of streets that might help understand the amplification/offsetting of the radiation - particularly the length scales in question.

4) The congruence with the quote by Weckwerth (1997) and subsequent sentences (line 210 - 217) really needs some deeper thought and analysis.

5) Please comment on how static heterogeneities might play out over land, where the 3-D solar radiation influence is significant. Particularly when the wind advects a boundary layer that includes the net effect of upstream static (and dynamic) heterogeneity. The scale of the patches and the advective wind will be important. This links in to my request to tie the discussion more tightly to the dynamic theory of streets.

6) Finally, the paper contains some testable hypotheses that I urge the authors to pursue with data since it will add much value to this line of research. (I'm not saying this should be done in the current paper.)

Minor:

7) Line 267: I think you mean "simulations" rather than data.

8) Line 272: Again please include more theoretical explanation of dynamically induced cloud streets.

9) When you use the phrase "surface heterogeneities" in the text, please be clear that this is a dynamical heterogeneity.

10) The LWP threshold > 0 for the cloud mask is much too rigid but I expect has little to no bearing on the results other than how it will bias the quoted cloud fractions. An optical depth threshold might be more useful/relevant anyhow.
* * *

---

## Author Comment (AC1) · 21 Aug 2017

Response to Anonymous referee #1

[Figure]

**1  General remarks**

First of all we wanted to thank you for taking your time to go through the manuscript in detail. Your contribution is very much appreciated. We appended a differential version of the manuscript. Before we proceed, we would like to clarify our intent of this study. The main purpose of the current manuscript is to show that the formation of cloud streets can be initiated with 3D radiative transfer alone. We want to stress that the first set of simulations that were conducted within this study are without any mean background wind or wind shear whatsoever. Of course, no wind at all is seldom the case in the atmosphere which leads naturally to the question how radiatively induced clouds streets compete or interact with dynamically induced organization. This second part of the study led to an interesting conclusion: While it was assumed that the radiative influence on the surface is smoothed by a horizontal wind, we find that the dynamical organization of clouds is capable to produce a static radiative pattern on the surface for an extended period of time, which in turn allows the radiation to change the flow.

We improved the manuscript to a) make the distinction between radiatively and dynamically aspects more clear and b) narrow down on the theory of cloud streets.

Answers to the specific comments are given below.

- *One of key aspect of the roll-type shallow convection is the presence of low-level shear associated with the Ekman boundary layer. This is really not mentioned in the introduction and I feel this is an essential omission. I think the shear explains the key impact of the mean wind as documented in Fig. 4. I have more points on that aspect in the specific comment section below, but a discussion of numerical studies (starting with Mason and Sykes QJ 1982, p. 801) as well as observational studies have to be brought in the revision (Weckwerth at el. is just mentioned in passing without any reference to the dynamics). There is also a wealth of theoretical studies on the stability of shear flows in unstable stratification focusing on*

*the development of roll-type circulations, starting with Asai (JMSJ 1970, p. 129). I understand that the authors specialize in the radiative transfer and not in the atmospheric dynamics, but the poor treatment of the dynamical aspects needs to be corrected. My suggestion is the authors trace back citations to the papers listed above and provide an appropriate discussion on the role of boundary-layer shear in determining the organization. Overall, I feel the dynamics is the key, and radiation provides just a small (although quite interesting!) modification. But I feel that unequivocally separate the two is difficult.*

Yes, literature separates the development of roll-vortex circulations into two regimes. Inflection point instabilities (associated with shear in an Ekman Boundary Layer) and thermal, buoyancy driven instabilities. To keep the simulations as simple as possible, we do not use a Coriolis force in the setup of the simulations and therefore do not have an Ekman boundary layer and the associated cross-wind shear. The formation of dynamically induced cloud streets in the here presented simulations are not explained by the cross-wind shear but rather stem from thermal instabilities. We therefore refrain from presenting details on inflection-point instability studies. That being said, it is clear that this reasoning and the description of the model setup needs further improvement and we hope that the revised manuscript clarifies on these points. We added a paragraph to the introductory part as well as to the description of the model and simulation setup.

- *The model setup is described with insufficient detail. For instance, sending the reader to the description of the land surface model in Heus et al is not appropriate. The $C_s kin$ parameter in Table 1 is not explained and I did not know what it really meant. In the discussion of model results this becomes obvious: this is the depth of the well-mixed layer of water that responds to radiative and surface heat fluxes. This is critical to the specifics of the simulation as the shadow on the surface is only important through its effect on the surface sensible and latent fluxes, doesn't*

[Figure]

*it? Ocean response to the shadow can be argued to be quite small at spatial and temporal scales this study is concerned with, whereas land surface would respond quite rapidly. Similarly, significant wind moves the cloud and its shadow, and the surface may not have time to respond. These aspects of the model need to be presented in detail so the reader is aware of the surface response in various simulations. Are the surface momentum fluxes (i.e., surface friction) included in the model setup? If so, what is the surface drag (or whatever parameter is used to describe surface roughness) for the momentum? For instance, to mimic the difference between land-surface (small $C_skin$) and the ocean (large $C_skin$), the surface drag should also be changed (larger over land and smaller over the ocean). This aspect should be at least mentioned in the description of the model as the surface drag affects the shear across the boundary layer.*

We improved the model description to introduce the skin heat capacity more pronounced and explain its role in our model setup.

And yes, you are correct, the radiative effects are stronger over land-surfaces and less so over ocean. This is one of the key results of this study. Concerning other changes in surface roughness and vegetation response etc., those are intendedly neglected with the goal to minimize the number of free parameters that might have an influence on the results. I agree that if we were to put up a realistic simulation on ocean and land-surface interactions, we would need to account for these differences but we concentrate mostly on a process understanding. We added a description and explanation that we keep them constant.

- *I would like to see more analysis of bulk properties of the cloud field to put model results into perspective. For instance, the authors should show evolution of the cloud cover for various simulations, BL depth (differences in the cloud mean size evident in Fig. 1 suggests to ma that BL is deeper in the upper panel as clouds seem larger), depth of the cloud field, wind profiles across BL in various simulations, etc. etc. Differences in those bulk properties can affect organization of*

[Figure]

*shallow convection as well and better isolating them from the effects of 3D radiation would be desirable. At the moment, the authors provide very speculative discussion of the model results (see specific comments) and I think some of the bulk differences may be used to better explain the results as well.*

Most bulk properties of the simulations develop very similar, irrespective of radiative transfer solver used and are primarily a function of the solar zenith angle which determines the total energy uptake of the simulations. The fact that the results give a clear signal for convective organization across the various zenith angles suggests that the mechanism is robust. Furthermore, much of the discussion compares simulations that only change the azimuth angle of the sun which exhibits exactly the same evolution of bulk properties and we feel that additional material in that respect would only elongate the paper. We added a reference to additional material concerning cloud fractions, liquid water paths and mean cloud size distributions.

- *As far as I can tell, shallow convection organization develops gradually and the time scale is relatively long (hours; this can also be better quantified in the analysis). In nature, the sun is moving around, so both the azimuth and zenith angles are slowly changing. So the idealized setup may be questioned if one has to wait long time for the organization to develop. This aspect needs at least to be recognized in the manuscript.*

I very much appreciate your comment and added a new figure to the manuscript that presents the timescales of the change in the convective organization. From the simulation results, we find that the organization due to radiation can happen in as little time as half an hour.

[Figure]

**2 Specific Comments:**

- *The title needs revision. "The role"..."on" is not correct. "In" would be better, but replacing "role" with "impact" would be more appropriate.*

  Changed to "in"

- *L. 92: "resolution" has to be replaced with "grid length".*

  Fixed

- *L. 96: I do not understand "...layers of the surface model are soaking (30Please rephrase. Is the Bowen ratio the same in all simulations? This affects buoyancy flux that drives the boundary layer dynamics.*

  We changed the wording of the sentence. The Bowen ratio primarily depends on the net energy uptake in a given simulation and changes in a subset of simulations. Regarding your question however, the moisture pool is sufficiently large as to not deplete over the course of the simulations. The Bowen Ratio is between $0.1$ and $0.5$.

- *L. 111. Lower sun means lower energy input, hence later convection development, correct? I would also think that this leads to different evolution of the boundary layer depth, an aspect that might be important as well.*

  That is correct. We hoped that we made a point that this parameter study aims to quantify the key mechanism of radiatively induced changes in convective organization. Particularly the discussion part of the manuscript aims to dissect the various influences of the parameters, including a paragraph specifically on the sensitivity of dynamically induced organization due to differing $Q_{net}$. Please see also the answer to a concern of reviewer #2.

- *I suggest adding a table with simulation acronyms and apply them throughout the text for an easy reference.*

I think this is a matter of taste and I personally always get confused when lots of acronyms are floating around. I had hoped that the recurring scheme of the five parameters sinks into the readers mind and with that one should be able to navigate the discussion.

- *Do various simulations have different destabilization rates across the lower troposphere? This may have some impact on convection as well. See major point 3 above.*

  I am not really sure what you mean by destabilization rate. If you presume that there is a background profile against which the simulations are nudged, there is none. Radiative tendencies are the only "external" forcing. In other words, the solar zenith angle in the simulations is the primary factor that will destabilize the atmosphere.

- *Fig. 2 is too small. Consider splitting into separate figures or use vertically-stack panels.*

  I increased the size of the panels and of the overlap and decreased the legends.

- *L. 136. Is this the wind direction, or shear? What is "along track"?*

  Wind direction, there is no cross-wind shear. I rephrased it to wind direction.

- *I feel Fig. 4 is the key result of the study. But some aspects are really not mentioned in the discussion: i) The spread between simulations with different $C_s kin$ narrows with the $C_s kin$ increase. Does this suggest some dynamical effects through surface fluxes? ii) The correlation ratio is much larger for the strong wind case, no doubt because of the role of Ekman shear across the boundary layer.*

  Indeed, Fig. 4 summarizes the parameter study.

  i) Yes, that is correct. The skin heat capacity controls the capability of radiation to create surface heterogeneities, the principal mechanism of radiatively induced

convective organization. There is a lengthy paragraph that explains this. I will try to rephrase to highlight it more.

ii) There is no Ekman shear because we do not employ coriolis forces. Stronger wind speeds with a linear wind shear profile does however induce stronger cloud streets. I hope this is clearer now in the revised manuscript.

- *I found the discussion in section 3 speculative and not supported by the analysis. For instance, Fig 5 can be supported by the analysis of model data. That said, my problem is that changes in the surface fluxes do not translate immediately into changes of the boundary-layer structure. The argument is likely correct for the surface layer, but I am not sure how rapidly these changes are passed higher up. Another aspect is the role of secondary circulations that can either support or suppress development of roll- type convection. The discussion on lines 185-190 seems to suggest that the authors think this happens, but I suggest using model data in an attempt to document that. For instance, are there any systematic differences in the updraft/downdraft structure between sunlit and shadow part of the cloud? One should investigate that.*

Regarding you first question: *If surface fluxes can penetrate cloud layer dynamics...*

Horn et al. 2015 investigate the timescales of surface heterogeneities and find changes up into the cloud layer. In contrast, Lohou and Patton, 2014 only find an impact of surface fluxes up to $0.2z_{BL}$ but they also state that this might be because of their strong horizontal background wind and the coupling might be more direct for smaller wind speeds. Finally, Gronemeier et al.(2016) for example, find indeed a coupling into the cloud layer, similar to ours.

This brings me already to the second part of your question: *Are there any systematic differences in the updraft/downdraft structure.*

This is a good question.... I would like to steer you towards Gronemeier et

al.(2016) who were able to average the wind field horizontally along one of the horizontal domain axis because they prescribed the surface heterogeneities. This allows to study the flow on the sunlit/shadowy sides. Our simulations do not have that rigorous symmetry. One may be able to track the clouds and analyze the wind field around individual clouds. That is, however, an involved task which, we feel, can not be part of this work.

- *How the wind (and thus the boundary layer shear) is maintained? Again, major point 2 above.*

  The simulations are started with an inital wind profile and inertia keeps it moving. The drag does not remove the wind on these short time scales.

- *L. 210 – 217. Can these speculations be supported by appropriate analysis of model data (e.g., shear, boundary layer depth, etc).*

  I think the theoretical foundations concerning as to where the limit of buoyancy vs. shear-stretching lies, are limited. Anyway, it is encouraging to see that the LES simulations reproduce the observations(Woodcock (1942); Priestley (1957); Grossman (1982)). We rewrote the paragraph.

- *Suggestion for the future: one can apply different surface roughness to explore the impact of shear. Also, one can vary Coriolis parameter (including a change of sign to mimic the southern hemisphere) to better separate dynamical and radiative effects.*

  Indeed, surface roughness or directly shear curvature could have been another parameter to include in the analysis. This might be a interesting application if one wants to understand the influence of atmospheric radiative heating rates.

- *L. 245-250: I am sure there are more recent references that show observational estimates of the relevant scales than Kuettner 1959.*

Probably, nowadays one can look for cloud street patterns in so many satellite pictures. Yet, the reference is still accurate and is a testimony for how long meteorologists have been fascinated by the occurrence of cloud streets.

- *I found the conclusion section too brief and not providing the justice to the wealth of results the authors have. In particular, dynamical aspects are really not discussed at the appropriate detail level throughout the text and thus in the summary section.*

A complete disentanglement of dynamical effects would be great and may be a topic for future studies. Here, we try to focus on radiative effects and particularly radiative surface impacts.

Many thanks,

Fabian Jakub

[Figure]

**Supplement:**

[revised manuscript text omitted]

---

## Author Comment (AC2) · 21 Aug 2017

Response to Anonymous referee #2

[Figure]

**1 General remarks**

First of all we wanted to thank you for taking your time to go through the manuscript in detail. Your contribution is very much appreciated. Answers to the specific comments are given below. We appended a differential version of the manuscript as supplement.

- *The paper is very short and could do with more material. To start with it should inform the reader about the theory of streets. Line 51/52 is insufficient. The authors tend to be in a hurry to tie the paper up and not deal with details like teasing out the extent to which horizontal photon transport contributes to the results (Line 190). I would have appreciated more analysis. A few choice simulations to focus on various issues would greatly add to the impact of the paper.*

  The introduction on the theory of cloud streets was also a concern for reviewer #1 and we added a paragraph to the introductory part as well as to the description of the model and simulation setup.

  We agree that it would be really interesting to study the effects of atmospheric heating. One could probably artificially increase the radiative heating rates and hopefully see a stronger signal in order to understand as to what extent and which mechanism is changing the cloud shapes. However, we do not think that the set of simulations with the chosen setup allows for further, quantitative analysis of the effects of atmospheric heating rates. The feedback through surface fluxes is most certainly the primary effect and has precedence in this study. We therefore added the study of atmospheric heating rates to the outlook of the paper.

- *The influence of 3-D longwave cooling should be discussed.*

  Indeed, we compute the thermal radiative transfer also in 3D but we expect the impact of 3D effects not to be important for the formation of cloud streets because thermal radiative transfer does not infer any asymmetries (i.e. is rotational symmetric). We added a paragraph to the model description.

- *I liked the intuitive sketch (Fig. 5) but would appreciate a similar sketch pertaining to the dynamics of streets that might help understand the amplification/offsetting of the radiation - particularly the length scales in question.*

We are not sure if we understand your request. If you mean a figure such as for example in Gronemeier et al. (2016), fig. 3, we feel that, in our case, it does not add a lot to the explanation pertaining the radiative/wind feedback. The length and time scales vary with zenith and azimuth angles and surface heat capacity and we could not come up with a simple sketch that would improve the display of our ideas.

- *The congruence with the quote by Weckwerth (1997) and subsequent sentences (line 210 - 217) really needs some deeper thought and analysis.*

I think the theoretical foundations concerning as to where the limit of buoyancy vs. shear-stretching lies, are limited. Anyway, it is encouraging to see that the LES simulations reproduce the observations (Woodcock (1942); Priestley (1957); Grossman (1982)). We rewrote the paragraph.

- *Please comment on how static heterogeneities might play out over land, where the 3- D solar radiation influence is significant. Particularly when the wind advects a boundary layer that includes the net effect of upstream static (and dynamic) heterogeneity. The scale of the patches and the advective wind will be important. This links in to my request to tie the discussion more tightly to the dynamic theory of streets.*

Static heterogeneities and their influence are in part tackled in Avissar and Schmidt (1998); Patton et al. (2005); Rieck et al. (2014). Furthermore, Gronemeier et al. (2016) investigated the interplay of static surface heterogeneities and radiatively induced, dynamic heterogeneities. While studying this interplay is clearly a very interesting and important aspect, we feel that there is probably not much potential gain in yet another study with idealized setups. We mention in

the outlook of the paper that we hope to study the effects of 3D radiative transfer in a more realistic setup within the High Definition Clouds and Precipitation for Climate Prediction (HD(CP)2) project.

- *Finally, the paper contains some testable hypotheses that I urge the authors to pursue with data since it will add much value to this line of research. (I'm not saying this should be done in the current paper.)* Thanks, I agree. Specifically, as noted in the point above, we very much look forward to checking whether we can reproduce the effects in a realistic setup and compare that to satellite observations. Another strategy we will try is to look for statistically significant organization of cloud streets in high resolution satellite imagery. Specifically whether the cloud streets follow the solar azimuth angles.

**2  Specific Comments:**

- *Line 267: I think you mean "simulations" rather than data.*
  Indeed, corrected.

- *Line 272: Again please include more theoretical explanation of dynamically induced cloud streets.*
  We added an additional paragraph to the introduction and rephrased this particular sentence.

- *When you use the phrase "surface heterogeneities" in the text, please be clear that this is a dynamical heterogeneity.*
  Yes, I went through the text and added clarifications where possible.

- *The LWP threshold $> 1$ for the cloud mask is much too rigid but I expect has little to no bearing on the results other than how it will bias the quoted cloud fractions. An optical depth threshold might be more useful/relevant anyhow.*

  Indeed, I checked and as you expected, it changes the cloud fraction usually by less than 1% and neither has an impact on the selection of time-steps nor on the autocorrelation ratios.

Many thanks,

Fabian Jakub

**Supplement:**

[revised manuscript text omitted]

---

## Author Response (AR2)

**Response to Anonymous referee #1**

- *It is still unclear what the (revised) description of the surface layer model humidity means ["soaking wet (30% water volume mixing ratio)"], lines 195-196. Is that what is sometimes referred to as surface moisture availability (e.g., 1 if the surface corresponds to the ocean and smaller value over land)? In a simple parameterization of the surface vapor flux, the surface temperature prescribes the water vapor mixing ratio as corresponding to saturation at the surface temperature and smaller a value over land. If I want to repeat the authors experiment I need to know what to use and currently I am left guessing. The explanation what $C_{skin}$ is still confusing, line 203, table 1. I looked at Hues et al. GMD 2010 and there I see $C_{sk}$ that is different to what the authors use in the paper under review. I think the units and values given in the table imply that $C_{skin}$ is simply the depth of a uniform-temperature layer of water for which the energy balance equation is given by $\rho_w c_w H dT/dt = net surface energy flux$. That is, $C_{skin}$ is the same as H in my formula. A complication is that Heus et al. also state that the surface model uses 4 layers. I think the authors need to be very specific what they use [1 layer, 4 layers? constant temp and moisture availability (or whatever variable is used)?]. Again, I would not be able to setup my own simulation following the authors' explanations.*

  I rewrote the explanation to further clarify the setup. We use four, saturated with respect to water, initially uniform layers, plus a skin layer on top with a varying heat capacity. Our $C_{skin}$ is the $C_{sk}$ of Heus et al. (2010). The latent heat release is solved by the land surface model in UCLA-LES, as described in Heus et al. (2010). Furthermore, for a full reference, I added the input files for the experiments to the code repository which is now mentioned in a Code-Availability section.

- *Additional analysis: I feel that the paper has been improved and it is close to be accepted. However, I still feel a couple details in the explanation of the model setup need to be corrected. I provide below a short list what I feel would be needed in association with Fig. 1. Figure 1 is quite impressive, but I wonder if the difference in cloud sizes (smaller in the lower panel to my eye) comes from difference in the boundary depth and/or radiative cooling (BTW, the radiative cooling is the "destabilization rate" in 5th bullet in specific comments section of my previous review; the authors were confused with what I meant). I feel that at least the lower-tropospheric moisture and temperature profiles, cloud fraction profiles, as well as mean radiative cooling should be shown together with the two panels. If they are really close, than please say so in the text to emphasize that other factors (like the BL depth) are practically the same in both simulations. But showing a figure would be better.*

  Yes, you are correct, the two simulations are not the same. The simulations start with the same initial profiles but then diverge. 3D radiative transfer illuminates the surface for an elongated period of time and tends to strengthen the updraft, leading to an invigorated growth of clouds. To compare our simulations with varying parameters, I think, we have two choices. One, we try to homogenize the forcing that drives the simulations (e.g. renormalize the surface fluxes to a fixed value) or, secondly, we don't evaluate the individual course of each simulation but rather look at the qualitative distribution of say, cloud streets.

I feel that the approach of homogenizing the fluxes would be rather difficult. Renormalized surface fluxes for simulations with large zenith angles for example would certainly introduce very high (physically unrealistic) peaks in illumination. I am not saying that these types of simulations are not interesting but I prefer to keep the interactions of the radiative transfer methods as physically sound as possible. While I think that it would be great if one could disentangle the various mechanisms (including the dependency on boundary layer height etc.), we feel that this study does already a great job in showing that there is a link between radiative transfer and the organization of convection through dynamic heterogeneities in surface fluxes. Irrespective of boundary layer characteristics, the fact that 1D simulations do not produce cloud streets, remains. Another example would be the fact that 3D simulations with a solar azimuth of $\phi = 90°$ and $\phi = 180°$ have the same boundary layer characteristics and cloud statistics but show the opposite sign in the autocorrelation ratio (i.e. orientation of streets). If you are interested in the evolution of individual simulations of shallow cumulus convection, pertaining 1D and 3D radiative transfer, maybe I can excite your interest in Jakub (2016, sec 3.2).

**Response to Anonymous referee #2**

- *I my first review I requested discussion of advected boundary layers. This was not included in the revised version. The advection of boundary layers adds significant complexity to the problem. Please add a few words in the closing paragraph (Moving forward...) along with an appropriate reference.*

  Indeed, the interplay of static and dynamic heterogeneities in conjunction with horizontal advection is a fascinating topic. We tried to keep the focus purely on the radiative effects (and as simple as possible) and did not delve into these aspects but we agree that it would be great to revisit earlier studies that tackled static heterogeneities in windy conditions. I added a paragraph to the outlook steering towards that line of research.

- *A technical editor should decide on orthogonal vs orthogonally and perpendicular vs. perpendicularly. I suspect it is the former in both cases.*

  I would appreciate that. I am not sure which is more appropriate either. I asked a native colleague and he suspects that the adverb form is the grammatically correct but that in case of perpendicular, he deems the adjective form to sound better.

- *Line 177 (marked version) change infer to cause*

  done.

- *Line 203 situations*

  done.

- *Line 204 a well mixed ocean*

  done.

- *Line 244 liquid*

  expanded.

- *Line 295 first develops*

  Indeed, what I meant was that first clouds develop after half an hour... I rewrote the sentence.

- *Caption figure 4: remove "Shown is"*

  Yes, done.

- *Line 441, remove ; and replace with )*

  The ; between citations is the default style. The ; at the end of the line in the *diff* version is apparently a hickup of latexdiff. It is fixed in the normal version.

- *Line 543 remove typically (the uncertainty is already conveyed by 'order of')*

  done.

[revised manuscript text omitted]